# Overview of Prognostic Systems for Hepatocellular Carcinoma and ITA.LI.CA External Validation of MESH and CNLC Classifications

**DOI:** 10.3390/cancers13071673

**Published:** 2021-04-02

**Authors:** Alessandro Vitale, Fabio Farinati, Michele Finotti, Chiara Di Renzo, Giuseppina Brancaccio, Fabio Piscaglia, Giuseppe Cabibbo, Eugenio Caturelli, Gabriele Missale, Fabio Marra, Rodolfo Sacco, Edoardo G. Giannini, Franco Trevisani, Umberto Cillo

**Affiliations:** 1Department of Surgery, Oncology and Gastroenterology, University of Padua, 35121 Padua, Italy; alessandro.vitale.10@gmail.com (A.V.); fabio.farinati@unipd.it (F.F.); germanclamp@outlook.com (C.D.R.); umberto.cillo@gmail.com (U.C.); 2Infectious Diseases Unit, Department of Internal Medicine, Padua University Hospital, 35123 Padua, Italy; ggbrancaccio@gmail.com; 3Division of Internal Medicine, Hepatobiliary and Immunoallergic Diseases, IRCCS Azienda Ospedaliero-Universitaria di Bologna, 40138 Bologna, Italy; fabio.piscaglia@unibo.it; 4Department of Medical and Surgical Sciences, Alma Mater Studiorum, University of Bologna, 40138 Bologna, Italy; 5Department of Health Promotion, Mother & Child Care, Internal Medicine & Medical Specialties, PROMISE, Gastroenterology & Hepatology Unit, University of Palermo, 90127 Palermo, Italy; g.cab@libero.it; 6Gastroenterology Unit, Belcolle Hospital, 01100 Viterbo, Italy; e.caturelli@tiscalinet.it; 7Infectious Diseases and Hepatology Unit, Azienda Ospedaliero-Universitaria of Parma, 43126 Parma, Italy; gabriele.missale@unipr.it; 8Internal Medicine and Hepatology Unit, Department of Experimental and Clinical Medicine, University of Firenze, 50139 Firenze, Italy; fabio.marra@unifi.it; 9Gastroenterology and Digestive Endoscopy Unit, Foggia University Hospital, 71122 Foggia, Italy; saccorodolfo@hotmail.com; 10Gastroenterology Unit, Department of Internal Medicine, University of Genoa, IRCCS Ospedale Policlinico San Martino, 16132 Genoa, Italy; egiannini@unige.it; 11Division of Semeiotics, IRCCS Azienda Ospedaliero-Universitaria di Bologna, 40138 Bologna, Italy; 12HCC Special Interest Group, Associazione Italiana per lo Studio del Fegato (AISF), 00199 Roma, Italy

**Keywords:** hepatocellular carcinoma, prognostic system, discrimination ability, homogeneity, monotonicity of gradients, prognostic performance

## Abstract

**Simple Summary:**

This review proposes a comprehensive overview of the main prognostic systems for HCC classified as prognostic scores, staging systems, or combined systems. Prognostic systems for HCC are usually compared in terms of homogeneity, monotonicity of gradients, and discrimination ability. However, despite the great number of published studies comparing HCC prognostic systems, it is rather difficult to identify a system that could be universally accepted as the best prognostic scheme for all HCC patients encountered in clinical practice. In order to give a contribute in this topic, we conducted a study aimed at externally validate the MESH score and the CNLC classification using the ITA.LI.CA database.

**Abstract:**

Prognostic assessment in patients with HCC remains an extremely difficult clinical task due to the complexity of this cancer where tumour characteristics interact with degree of liver dysfunction, patient general health status, and a large span of available treatment options. Several prognostic systems have been proposed in the last three decades, both from the Asian and European/North American countries. Prognostic scores, such as the CLIP score and the recent MESH score, have been generated on a solid statistical basis from real life population data, while staging systems, such as the BCLC scheme and the recent CNLC classification, have been created by experts according to recent HCC prognostic evidences from the literature. A third category includes combined prognostic systems that can be used both as prognostic scores and staging systems. A recent example is the ITA.LI.CA prognostic system including either a prognostic score and a simplified staging system. This review focuses first on an overview of the main prognostic systems for HCC classified according to the above three categories, and, second, on a comprehensive description of the methodology required for a correct comparison between different systems in terms of prognostic performance. In this second section the main studies in the literature comparing different prognostic systems are described in detail. Lastly, a formal comparison between the last prognostic systems proposed for each of the above three categories is performed using a large Italian database including 6882 HCC patients in order to concretely apply the comparison rules previously described.

## 1. Introduction

Ideal staging systems and prognostic scores for cancer management should offer a common scale to provide an accurate prognostic prediction for specific populations as well as for individual patients (“precision medicine”), appropriate selection criteria for the treatment avoiding under- and over-treatment, and an optimal design of randomized controlled trials. Conventional staging systems consider only the morphologic features of the tumour, while prognostic scores usually take into account all the main aspects affecting the final prognosis. Staging system and prognostic score should be easy to use, reproducible and transportable to different populations in order to be recommended and used on a large scale [1].

Unlike other tumours, hepatocellular carcinoma (HCC) usually arises in the context of another remarkable disease, i.e., liver cirrhosis, making its management unique and more complex with respect to other malignancies. The prognosis of HCC patient is related to three main factors [2]: tumour burden and aggressiveness, liver dysfunction degree and the general health status of the patient. Number and size of lesions, vascular invasion and metastatic spread usually define tumour burden. Alpha-fetoprotein (aFP) level has been included in some prognostic systems to describe tumour aggressiveness. Liver function is usually evaluated through biochemical markers (albumin, bilirubin, prothrombin time) and signs and symptoms of liver dysfunction (ascites, encephalopathy, portal hypertension, impaired renal function, hyponatremia), or through the inclusion of multiparametric liver function scores such as the Model for End Stage Liver Disease (MELD) score [3], Child-Pugh score (CPS) [4] or albumin-bilirubin (ALBI) score [5]. The Eastern Cooperative Oncology Group (ECOG) Perfomance Status (PS) [6] or the Karnofsky index [7] are used to describe the general health of the patient.

During the last three decades several staging systems or prognostic scores have been proposed from both the Asian and European/North American world to estimate the prognosis of HCC patients (Figure 1). We can classify these systems/scores into three main categories, based on the methodology by which they were created:(1)Prognostic scores, derived from real cohort populations.(2)Staging systems, derived from the literature review(3)Combined prognostic systems, based on the literature evidences but weighted in a real population, and with the possibility to be used both as scores and as staging systems.

In this paper, we first present an overview of the main prognostic systems classified according the above three categories for the general population of HCC and, hence, not for specific sub-populations of HCC patients (i.e., early, intermediate, or advanced HCC) or only for specific treatments (i.e., liver transplantation, liver resection, ablation, intra-arterial therapies, or systemic therapies).

A second section is dedicated to a comprehensive description of the methodology required for a correct comparison between different systems in terms of prognostic performance. In this section we also describe the main studies comparing different prognostic systems with the aim of identifying the one with the best performance.

Third, in order to concretely apply the comparison rules previously described to the need of clinical practice, we have performed a formal comparison between the most recent prognostic systems proposed for each of the above mentioned three categories, using a large Italian database. In particular, we have compared the Taiwanese Model to Estimate Survival for HCC (MESH) [8] score for the prognostic scores category, the Chinese Liver Cancer (CNLC) classification [9,10,11] for the staging systems category, and the Italian Liver Cancer (ITA.LI.CA) prognostic score and staging system [12] for the combined prognostic systems category. The Cancer of the Liver Italian Program score [13,14] and the Barcelona Clinic Liver Cancer (BCLC) classification [15] are also taken as reference for the prognostic scores and the staging systems categories, respectively. From this point of view, this study represents also an external validation of the MESH and CNLC prognostic systems.

## 2. Overview of Available Prognostic Systems for HCC

### 2.1. Prognostic Scores

Table 1 describes all the most important data-based prognostic scores, whereas, in this paragraph we analyse only the four main ones: the Okuda system [16], the CLIP score [14], the Japanese Integrated Staging score [17,18] and the MESH score [8].

The Okuda staging system, proposed in 1984, represents the first attempt to stage HCC including variables aimed at weighting the contribution of cirrhosis to the patient prognosis [16]. It indeed combines the anatomical extension of the tumour (≤ or >50% involvement of the liver) to the liver dysfunction (expressed by albumin, bilirubin, presence of ascites). Nowadays, the Okuda system has been progressively abandoned as its main limit is the dichotomous vision of HCC size, that makes this system not useful in modern clinical practice where a considerable percentage of HCC are detected before their burden crosses the 50% of the liver volume.

The CLIP score [13,14] was developed through a retrospective cohort study and has been considered an excellent prognostic score, especially because it has been externally validated. Unfortunately, it does not consider the patient clinical status and it is scarcely sensitive in stratifying early HCCs, amenable to curative treatments such as percutaneous ablation or surgical therapies.

Japanese HCC experts proposed the JIS score [18] that combines the Japanese TNM and Child-Pugh (C-P) classifications. This score lacks a strong external validation in European/North American countries, and it is almost exclusively used in Japan.

The MESH score [8] is the last proposed data-based prognostic score for HCC (Table 2) which was assembled using data of 3182 prospectively enrolled patients. This score (ranging from 0 to 6 points) combines Milan Criteria, presence and type of vascular invasion, C-P score, performance status and laboratory parameters (aFP and Alkaline Phosphatase). Even this system does not propose treatment recommendations. However, MESH score had at least on external validation in European/North American countries [19].

### 2.2. Staging Systems

TNM, BCLC and CNLC staging systems are the main examples of evidence-based systems.

As for other cancers, the TNM system [24] is based on tumour pathological features, but it does not consider the liver function and does not stratify for the patient general health condition.

The BCLC classification [15], proposed in 1999, was the first system integrating liver function assessment, tumour extension and also patient general health status. It classifies patients into five subgroups, from 0 to D, and each group is associated with a specific therapy. This classification can be considered an evidence-based system, since it was generated by analysing the results of randomized controlled studies testing a given treatment versus placebo in patients with comparable tumour characteristics and liver function. The BCLC system has been endorsed by the American Association for the Study of Liver Diseases (AASLD), the American Gastroenterology Association (AGA), the European Association for the Study of Liver (EASL), and the European Organization for Research and Treatment of Cancer (EORTC) [25,26].

Over the years, the BCLC flow chart has been frequently modified. It is not an aim of this study to discuss pros and cons of the BCLC treatment algorithm [27]. The BCLC suffers from the fact that it was not created and weighted in “real-world” HCC populations. As a result, its prognostic performance is usually lower than that of data-based prognostic scores [28,29]. In addition, some potential limits of the BCLC structure that could affect the prognostic power of this system are: (a) the absence of a size cut-off for single HCC in early stage; (b) the high heterogeneity of intermediate and advanced stages; (c) the absence of a clear distinction between intra- and extra-hepatic vascular invasion; (d) the absence of prognostic biomarkers such as aFP; (e) the excessive prognostic weight given to performance status 1; (f) the poor prognostic stratification of liver dysfunction degree (i.e., only a simple distinction between Child-Pugh C and Child A-B classes is proposed in the original BCLC scheme).

Finally, the CNLC staging system [9,10] represents the chart endorsed by the National Health and Family Planning Commission of the People’s Republic of China for HCC surveillance, diagnosis, staging and treatment (Table 3). These recommendations, released in 2017 and updated in 2019, are a management summary regarding all the aspect of HCC patients delineated by a multidisciplinary panel of more than 50 experts, including surgeons, oncologists, hepatologists, interventional radiologists etc. This staging system takes into account patient general health status, tumour burden and liver function. It has several similarities with the BCLC system, but it supports more aggressive treatment options for advanced HCC stages. For example, the CNCL system indicates liver resection in patients belonging to Ia, Ib, and IIa categories, that correspond to the BCLC B stage with 2–3 nodules >3 cm, and also for select patients classified in IIb and IIIa stage (multinodular and locally advanced HCC) [9,10,11]. The updated 2019 CNCL version [11] is reported in the Table 3.

### 2.3. Combined Staging Systems

The Hong Kong Liver Cancer (HKLC) [30] and ITA.LI.CA [12] prognostic systems are the two main examples of combined systems.

The HKLC [30] was developed in 2014, predominantly on a cohort of patients with HBV-related HCC. In this score performance status, C-P score, tumour status (based on Milan Criteria), intra- and extra-hepatic vascular invasion or metastases were the pre-defined criteria, based on literature evidence. These variables were subsequently weighted in a real population in order to assign a relative coefficient to each of them.

The HKLC system can be used both as a prognostic score and as a staging system to help treatment assignment.

The HKLC, compared to BCLC classification, has better ability to prognostically stratify patients assigned to BCLC intermediate and advanced stages, who can therefore benefit from more aggressive treatments than those recommended by the BCLC system. The pitfall of HKLC system is the lack of solid external validation in a non-Asian population. Very few studies have compared BCLC and HKLC scores, and in European/North American populations the latter did not show a better prognostic performance than BCLC [23,30,31,32].

The ITA.LI.CA prognostic system [12], created in 2016 through a multicentre retrospective analysis and validated in a Taiwanese cohort, is a prognostic model able to efficiently predict the outcomes of HCC patients. It can be used as a prognostic score based on tumour burden, liver function and other patient-related variables (Table 4 and Table 5). This system recalls the BCLC classification concerning the stratification of tumour characteristics in different stages, but with provides a better definition of the intermediate stage based on literature evidences. In particular, the intermediate stage has been arranged in three sub-groups. A size cut-off was introduced for single tumour to distinguish between stage A and B1. Furthermore, intra and extra-hepatic HCC vascular invasion were identified as separate entities, also considering that HCC with intra-hepatic vascular invasion is liable of therapeutic options with radical intent [12]. Patient functional status was evaluated with the C-P score and the ECOG performance staus. Lastly, aFP, which provides important prognostic information, has been added.

Each variable showed a different impact in determining the final score, and, consequently, different points were attributed to variables in order to correctly weight their prognostic influence. Lastly, based on these scheme, the ITA.LI.CA integrated prognostic score has been created. The lowest score (score 0) of the model corresponds to the best prognosis, while the highest one (score 13) depicts the worst prognostic scenario.

In the original study by Farinati et al. [12], this score was internally and externally validated in a large Taiwanese cohort, and more recently, Borzio et al. [33] externally validated the ITA.LI.CA score in an independent multicenter cohort study including 1508 HCC patients. The ITA.LI.CA score has been found to perform better than other scores even in restaging patients at the time of HCC recurrence and before treatment decisions [34].

In conclusion, ITA.LI.CA showed a great ability to predict the prognosis in HCC patients. Moreover, the ITA.LI.CA prognostic system can be also converted in a simple ITA.LI.CA staging to assist treatment allocation [35]. This innovative staging system proposes therapeutic options for each stage based on the so called “treatment hierarchy”, an approach inspired by the Precision Medicine alternative to the “stage hierarchy” concept [34,36].

### 2.4. Summary of the Pros and Cons of Prognostic Systems

Prognostic scores are usually developed from a real-life cohort population using objective and reproducible variables. These systems rely on a rigorous statistical methodology usually based on multivariable survival models derived from a process that is agnostic to known risk factors. This peculiar statistical process explains why these score have often a good prognostic performance. Unfortunately, not all these systems have been internally and externally validated. Moreover, since they are developed from a specific population, their application to the general population is not always feasible and, even most importantly, they do not define tumor stages able to guide the treatment selection.

Staging systems are usually created by a panel of experts who establish different prognostic stages based on the evidence of the scientific literature. The main advantage of these systems is that they offer a potential linkage between HCC stage and treatment. However, they are based on a weak statistical methodology, and, for this reason, they usually show a lower prognostic power than prognostic scores.

Combined staging systems are developed from evidence-based composite variables (i.e., Child-Pugh score, tumour features) a priori defined by experts. These composite variables are then weighted in a real population to create the prognostic score. The score is usually also converted to a staging system to allow and facilitate treatment assignment. The theoretical advantage of combined systems is that they allow obtaining, simultaneously and in a balanced manner, a good prognostic evaluation (using the prognostic score) and an appropriate treatment allocation (using the staging system). These features theoretically make combined staging systems more effective and clinically useful than prognostic scores and staging systems categories.

A relevant issue for HCC clinical management is the relationship between prognostic systems and treatment choice [36]_._ This complex relationship can be analyzed from two points of view. The first is mainly a prognostic point of view. Since, treatment selection is influenced by different prognostic variables (i.e., tumour characteristics, liver function, and patient general conditions) there is a statistical interaction between treatment and other variables, so treatment can not be included as an additive variable in a general prognostic system. From this specific prognostic point of view, therefore, commonly used prognostic systems (described in this paper) can be used for a prognostic assessment for the general HCC population, but specific prognostic scores for each treatment should be used to obtain a more accurate prognostic estimation after that treatment decision is taken [34]. In this review we only described prognostic systems designed for a general HCC population independently from treatment choice, while treatment specific prognostic scores are not object of this study.

The second point concerns the relationship between prognostic systems and treatment assignment. As described in this paper, only staging and combined systems categories proposed treatment algorithms for HCC patients. Several evidences from the literature showed, however, that adherence to these algorithms (i.e., linking treatment choice to a specific stage according to the stage hierarchy philosophy) was very low in every day clinical practice [31,37,38,39,40]. A multidisciplinary evaluation aimed to obtain a personalized treatment decision is probably the best way to optimize HCC patient outcome. On this perspective, the treatment hierarchy approach is closer than stage hierarchy to precision medicine therapeutic approach for HCC [36].

## 3. Comparison of Available Prognostic Systems

The performance of a prognostic system is defined by three characteristics: homogeneity, discriminatory ability and monotonicity of gradients [41]. A system is homogeneous when differences in survival between patients of the same stage are small. The discriminatory power is the ability of the system to produce great differences in survival among patients in different stages. When monotonicity of gradients is fulfilled, the survival of patients in each stage is longer than that of patients in the subsequent adjacent stage.

These three characteristics are measured using the likelihood ratio (LR) derived by a Cox regression model, the Akaike Information Criterion (AIC), the Harrell’s C-index, and the *X*^2^ linear trend test (LT). The AIC is calculated from the LR test and it is particularly useful to compare ordinary prognostic systems with a different number of stages/points. A low AIC value (corresponding to a high LR test) testifies a high homogeneity and monotonicity of gradients, while high values of C-index and LT test indicate high discriminatory ability and monotonicity of gradients [42,43].

A lot of comparative studies have been conducted with the goal to identify the system with the best prognostic power in HCC patients. In the study of Marrero et al. [44], the BCLC staging system, as compared with six other prognostication systems, showed the best independent predictive power for survival. The superiority of BCLC staging system was supported by external validations in Korean [45] and Italian populations [46]. However, in the last years, inherent limitations of the BCLC system have emerged. In 2016, Liu et al. [47] compared 11 staging systems in a large prospective database including 3182 HCC patients. The ability to predict the prognosis was analysed through the homogeneity and corrected AICc. This study obtained low AICs for the BCLC system, while the CLIP system resulted to be the best prognostic model in all patients as well as in the subsets created according to the aetiology and treatment strategy. Therefore, this study showed that a data-based score (CLIP score) performs better than an evidence-based system.

The study by Farinati et al. [12] assessed the prognostic powers of the ITA.LI.CA, BCLC, HKLC, MESIAH, CLIP, and JIS systems. The ITA.LI.CA score showed the best discriminatory ability and monotonicity of gradients in all three study cohorts (training, internal validation and external validation). In particular, the C-index of the ITA.LI.CA score was 0.71 and 0.78 in the internal and external validation cohort, respectively. The LR test indicated that the ITA.LI.CA system had a significantly better discrimination ability (*p <* 0.001) than the other systems in all three studied groups, and the superiority of the ITA.LI.CA score was also confirmed after stratification for time-period. The ITA.LI.CA prognostic system shows a great ability to predict individual survival in European and Asian populations [12,48].

However, it is worth to note that the prognostic ability depends on several variables, including time period, the geographical location of the study, numbers and type of patient population, modality of comparison and type of HCC treatment(s) mainly adopted in the analyzed population (Table 6). Indeed, therapeutic management can greatly affect the predictive power of a score, so that the best staging system for HCC patient undergoing liver resection might not be the same of that showing the best performances in patients receiving palliative treatments or supportive care. As a matter of fact, in a Taiwanese cohort of 2010 patients, the survival was better predicted by the Tokyo staging system in patients undergoing liver resection, and by the CLIP score in patients who received chemotherapy or supportive care [49].

The geographical area, Asian or European/North American region, can also influence the staging system performance throughout a number of factors, including etiology of liver disease, tumor biology, predominant stage(s) at the time of HCC diagnosis and treatment strategies [50]. In particular, in Asian the TNM, JIS and CLIP systems showed the best predictive power, while in the European/North American countries BCLC and CLIP showed the best discriminatory power. Therefore, it is easy to understand why no universal consensus has been so far reached on which prognostic system can be considered the best one. [46,47,48,49,50,51].

## 4. External Validation of MESH and CNLC Prognostic Systems in the ITA.LI.CA Database

### 4.1. Study Population

With the intent to compare in real life clinical practice the prognostic performance of more recent HCC prognostic systems, we used the last version of the Italian Liver Cancer (ITA.LI.CA) database. This database is a large, multi-centre registry containing prospectively collected data of patients with a newly diagnosed and recurrent HCC managed in 23 Italian centers with different levels of expertise (secondary and tertiary referral centres) [77,78]. It currently includes 7816 HCC patients consecutively evaluated and managed from January 1987 to December 2018, and its data are updated every 2 years and periodically revised by the coordinator center (Semeiotics Unit, Alma Mater Studiorum-Bologna University). The management of the ITA.LI.CA database conforms to the Italian legislation on privacy. According to the Italian laws, no specific patient approval is needed for any retrospective analysis, but all patients provided written informed consent for every diagnostic and therapeutic procedure, as well as for having their clinical data recorded anonymously in the ITA.LI.CA database. The study was approved by the Institutional Review Board of the ITA.LI.CA coordinating center, Alma Mater Studiorum University of Bologna (approval number 99/2012/O/Oss), and it was conducted in accordance to the ethical guidelines of the 1975 Declaration of Helsinki.

In order to limit the potential bias to include patients managed with old imaging and treatment tools, we only enrolled 6882 patients with their HCC diagnosed between 2000 and 2018.

This database, due to its heterogeneity in terms of tumour stage, underlying liver disease severity, and therapeutic approaches, provides a reliable insight into the characteristics of HCC in a European/North American population and, therefore, it represents an optimal substrate for the external validation of the new MESH [8] and CNLC [9,10,11] scores in real-life clinical practice.

Baseline characteristics of the study cohort are described in the Appendix A. Enrolled patients were classified according to ITA.LI.CA prognostic score [12], ITA.LI.CA simplified staging [35], CLIP [14] and MESH [8] scores, and BCLC [15] and CNCL [9,10,11] staging systems (Appendix A).

### 4.2. Statistical Analysis

Baseline characteristics were examined based on frequency distribution; continuous data are presented as median (interquartile range) unless otherwise indicated.

Overall survival was defined as the time elapsed from the date of HCC diagnosis to the date of death, last follow-up evaluation, or data censoring (31 December 2019). Kaplan- Meier survival curves were used to estimate the median overall survival (OS) and 1-, 3-, 5-, and 10-year survival rates. Kaplan Meier curves were used to describe survival figures of different stages of each prognostic system, and the log rank test was used to compare differences in survival. As detailed in Section 3 of this article, we compared the prognostic performance of different HCC prognostic systems in terms homogeneity, monotonicity of gradients, and discrimination ability using the LT chi-square, the AIC value and the C-index tests. In addition, the prognostic systems with the best prognostic performance was compared with the other systems by using the likelihood ratio test (the higher the test value, the greater the superiority). Missing data of study covariates always involved <10% of patients. Thus, they were estimated using the Maximum Likelihood Estimation method [79]. A two-tailed *p*-value < 0.05 was considered statistically significant, and analyses were performed with JMP^®^ Pro 15.2.0 package (2019 SAS Institute Inc., 100 SAS Campus Drive Cary, NC 27513-2414, USA), STATA13.0 (Copyright 1985–2013 StataCorp LP, 4905 Lakeway Drive College Station, TX 77845-4512, USA) and R. app 4.0.0 GUI 1.71 (S. Urbanek and H.-J. Bibiko, © R Foundation for Statistical Computing, Wirtschaftsuniversität Wien, Welthandelsplatz 1, 1020 Vienna, Austria 2016).

### 4.3. Survival Analysis and Comparison between HCC Prognostic Systems

Median duration of follow-up was 79 months (interquartile range, 46–119 months). Median OS was 32 months (12–79 months), and the 1-, 3-, and 5-year survival rates were 75%, 48%, and 33%, respectively.

All the prognostic systems showed good discrimination ability at the first evaluation based on Kaplan Meier survival figures (Figure 2). Median OS (95% confidence intervals) of different stages/scores are described in detail in Appendix A.

Evidence-based systems (BCLC, CNLC) and the ITA.LI.CA simplified staging showed some limitations in the discrimination ability for advanced stages (Figure 2), namely, stages C and D for BCLC, stages IIIa, IIIb, and IV for CNLC, and stages C and D for ITA.LI.CA simplified staging.

Recently proposed prognostic models, such as MESH score and CNLC classification, showed better homogeneity, discriminatory ability and monotonicity of gradients than the older BCLC staging and CLIP score (Table 7). Nevertheless, the ITA.LI.CA score and the ITA.LI.CA simplified staging showed the best prognostic performance among evaluated systems. In particular, the C statistic of the ITA.LI.CA score in the whole study group was 0.693, a value superior to that of the ITA.LI.CA simplified staging (0.667), MESH (0.662), CNLC (0.661), BCLC (0.659) and CLIP (0.620). According to the likelihood ratio test, the prognostic performance of the ITA.LI.CA score was, once again, better than that of the other systems (*p* < 0.0001).

## 5. Conclusions

This review proposes a comprehensive overview of the main prognostic systems for HCC. Data-based prognostic scores, such as the CLIP score and the recent MESH score, being created on a solid statistical basis, generally have a good prognostic performance. However, for the same reason, they show a good prognostic performance in populations similar to the one in which they were generated, while the performance worsens in geographically and ethnically different cohorts. Moreover, they are of limited utility in supporting the therapeutic choice given their intrinsic “score” structure.

Evidence-based staging systems, such as the BCLC system and the recent CNLC classification, are useful in assisting treatment selection since they are usually created as treatment algorithms. However, since their structural variables are not prognostically weighted in real life populations, they often have a prognostic performance lower than that of genuine prognostic scores. Moreover, they carry the risk to limit personalized treatment of HCC patients strictly linking a given treatment to a specific stage (“stage hierarchy” approach) [36].

Combined prognostic systems are created from evidence-based simple or integrated variables (i.e., Child-Pugh score, tumour features, ECOG performance status, biomarkers such as aFP) that are prognostically weighted in real populations. The main examples of these models are the HKLC and the ITA.LI.CA prognostic systems. These systems have the potentiality to guarantee a good prognostic performance coupled with the ability to help in the treatment choice. Nonetheless, so far they are still very seldom used in clinical practice.

Prognostic systems for HCC are usually compared in terms of homogeneity, monotonicity of gradients, and discrimination ability. However, despite the great number of published studies comparing HCC staging/scoring systems, it is rather difficult to identify a system that could be universally accepted as the best prognostic scheme for all HCC patients encountered in clinical practice. We conducted a study aimed at externally validate the MESH score and the CNLC classification using the ITA.LI.CA database.

These two new systems confirmed a good homogeneity, monotonicity of gradients and discrimination ability also in our large Western HCC population. However, their performance was inferior to that of the ITA.LI.CA score and the ITA.LI.CA simplified staging. Nevertheless, it should be consider that this inferiority could be, at least in part, due to the fact that the comparison was made in the population from which the ITALICA prognostic model have been generated.

The results of our comparison, however, suggest some conclusions as far as survival prediction of HCC patients is concerned. First, modern prognostic systems seem to perform better for HCC patients than the older ones of the same category (i.e., MESH works better than CLIP score among prognostic scores; CNLC works better than BCLC among staging systems). Second, prognostic scores seem to perform better than staging systems. Third, all currently available prognostic systems have a suboptimal prognostic performance (C index ≤ 0.7), suggesting that substantial improvements are needed. The inclusion within these systems of biological markers measuring the cancer aggressiveness is likely the key factor that will allow to reach an optimal prognostic estimation of HCC patients survival outcome.

## Figures and Tables

**Figure 1 cancers-13-01673-f001:**
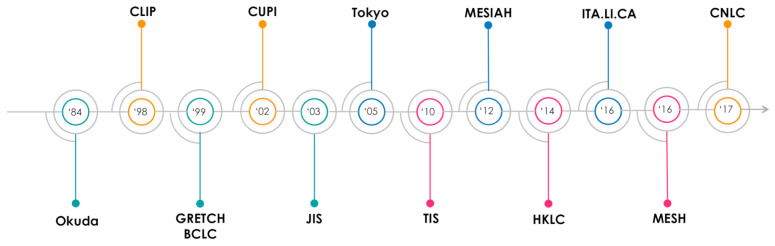
Hepatocellular carcinoma integrated prognostic systems proposed over time.

**Figure 2 cancers-13-01673-f002:**
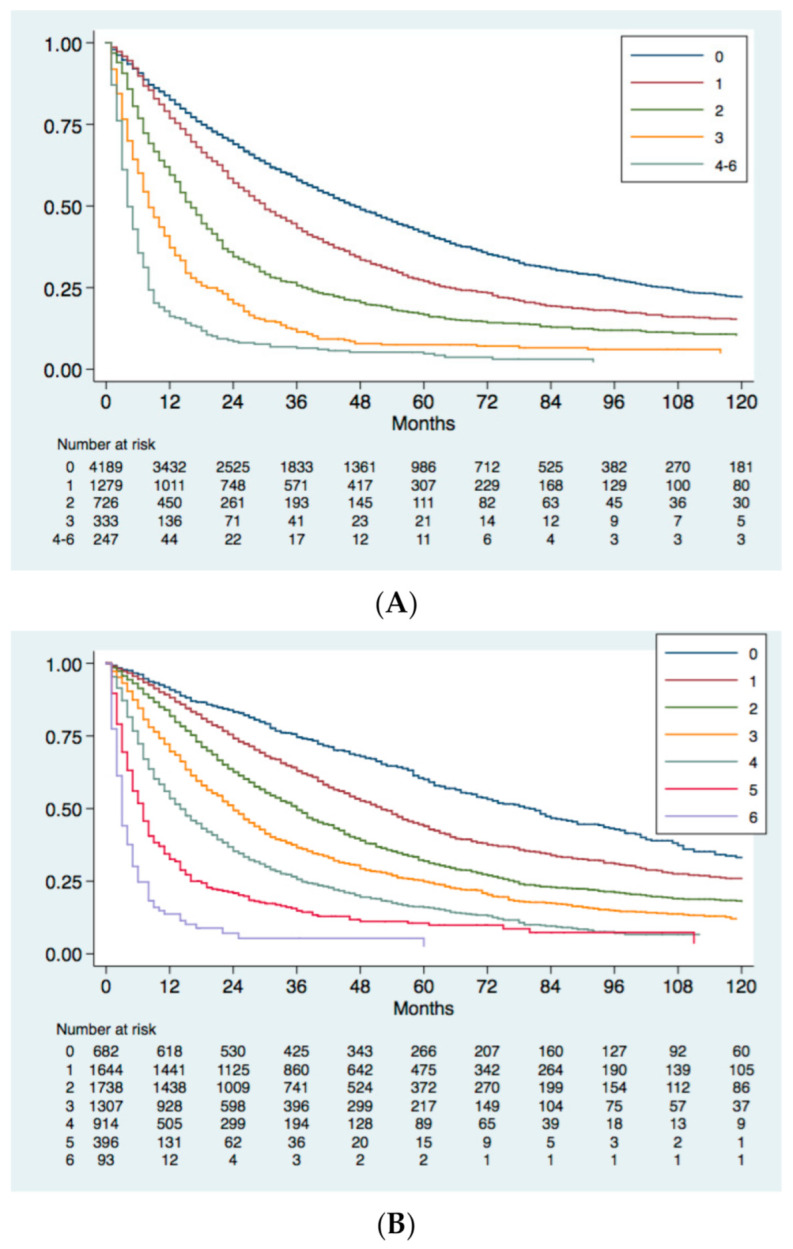
Kaplan-Meier survival curve of the study group stratified according to CLIP (**A**) and MESH (**B**) scores, BCLC (**C**) and CNLC (**D**) stages, ITA.LI.CA score (**E**) and ITA.LI.CA simplified staging (**F**). Log-rank test was always *p* < 0.001.

**Table 1 cancers-13-01673-t001:** Main HCC data based scores and their associated variables.

Staging System	Year	Pts Number	Performance Status	Liver Function	HCC Number	Size	aFP	Vascular Invasion	Metastasis	Other
**Okuda** [16]	1984	600	No	Ascites Albumin Bilirubin	No	Yes	No	No	No	/
**CLIP** [14]	1998	435	No	CPS	Yes	Yes	Yes	Yes	No	/
**French** **(GRETCH)** [20]	1999	761	Karnofsky	Bilirubin	No	No	Yes	Yes	No	Alk-P
**CUPI** [21]	2002	926	Symptoms	Ascites Bilirubin	Yes	Yes	Yes	Yes	Yes	Alk-P
**JIS** [18]	2003	Review	No	CPS	Yes	Yes	No	Yes	Yes	/
**Tokyo** [17]	2005	403	No	Albumin Bilirubin	Yes	Yes	No	No	No	/
**TIS** [22]	2010	2030	No	CPS	TTV	TTV	Yes	No	No	/
**MESIAH** [23]	2012	477	No	MELD Albumin	Yes	Yes	Yes	Yes	Yes	Age
**Taiwanese** **MESH** [8]	2016	3182	ECOG	CPS	Yes	Yes	Yes	Yes	Yes	Alk-P

aFP: alpha fetoprotein; Alk-P: phosphatase alkaline; CLIP: Cancer of the Liver Italian Program; CPS: Child Pugh Score; GRETCH: GRoupe d’Etude et de Traitement du Carcinome Hépatocellulaire; MELD: Mayo End stage Liver Disease; MESH: Model to Estimate Survival for HCC; MESIAH: Model to Estimate Survival in Ambulatory HCC; MC: Milan Criteria; JIS Japanese Integrated Staging; PVT: Portal Vein Thrombosis; TIS: Taipei Integrated Scoring System.

**Table 2 cancers-13-01673-t002:** MESH score and its associated variables.

Prognostic Factors	Scores	
	0	1
Tumour burden	Within MC	Beyond MC
Vascular invasion or metastases	Absent	Present
CPS score	5	≥6
PS	0–1	≥2
Serum aFP	<20 ng/mL	≥20 ng/mL
Serum Alk-P	<200 IU/L	≥200 IU/L

aFP: alpha fetoprotein; Alk-P: phosphatase alkaline; CPS: Child Pugh Score; PS: Performance Status.

**Table 3 cancers-13-01673-t003:** HCC staging according to CNCL staging system (2019 version) [11].

Variables	CNCL Stage						
	Ia	Ib	IIa	IIb	IIIa	IIIb	IV
PS	0–2	0–2	0–2	0–2	0–2	0–2	3–4
CPS	A-B	A-B	A-B	A-B	A-B	A-B	C
Number and size of HCC	Single ≤ 5 cm	Single HCC > 5 cm OR 2–3 HCC ≤ 3 cm	2–3 HCC >3 cm	≥4, no size limits	No number or size limits	No number or size limits	No number or size limits
Vascular invasion	No	No	No	No	Yes	Yes	Yes/No
Extrahepatic metastases	No	No	No	No	No	Yes	Yes/No

CPS: Child Pugh Score; HCC: Hepatocellular Carcinoma; PS: Performance Status.

**Table 4 cancers-13-01673-t004:** ITA.LI.CA prognostic score.

SCORE	0	1		2		3		4		5
*Tumor Stage*	0	A		B1		B2		B3		C
Diameter (cm)	<2	≤3	2–5	≤5	>5	>5	≤5	>5	Any	Any
Number of Nodules	1	2–3	1	2–3	1	2–3	>3	>3	Any	Any
HVI and/or metastases	No	No	No	No	No	No	No	No	Intra HVI	Extra HVI or Metastases
*Functional Score*										
CPS	5	6		7		8		9		10–15
ECOG PS	0	1		2						3–4
*aFP*	≤1000 ng/mL					>1000 ng/mL				

aFP: Alfa FetoProtein; Child Pugh Score; ECOG PS: Eastern Cooperative Oncology Group Performance Status; HVI: Hepatic Vascular invasion.

**Table 5 cancers-13-01673-t005:** ITA.LI.CA simplified staging for treatment allocation [35].

**Tumor Stage**	**Diameter (cm)**	<2	≤3	≤5	3–5	>5	≤5	>5	>5	Any	Any	Any
**Number of Nodules**	1	2–3	1	2–3	1	>3	2–3	>3	Any	Any	Any
**HVI and/or Metastases**	No	No	No	No	No	No	No	No	Intra HVI	Extra HVI or Metastases	Any
**Functional Score**	CPS ≤9 and PS 0or CPS ≤7 and PS 1–2	CPS 8–9 and PS 1–2, or CPS >9, or PS >2
**Staging**	**0**	**A**	**B1**	**B2**	**B3**	**C**	**D**
**Therapy**	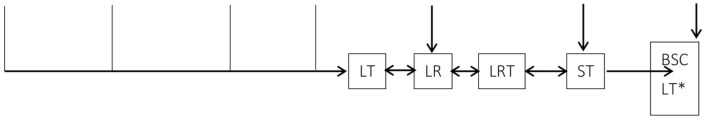

* LT if no Intrahepatic vascular invasion or <3 nodules with largest < 5 cm OR aFP < 1000 ng/mL. LT: Liver Transplantation; LR: Liver Resection; LRT: Loco Regional Therapy; ST: Systematic Therapy.

**Table 6 cancers-13-01673-t006:** Comparison of the prognostic systems in the literature using the LR ratio and AIC index.

Authors	Year	Country	Study	*n*. of Patients	Median F.U. Months	Modality Comparison	Compared Staging Systems	Most Representative Stages	Results of Comparison		Conclusions
Ueno et al. [52]	‘00	Japan	R	662	-	LR	CLIP	1 (195); 2 (169)	184.34	-	The CLIP score has the highest stratification ability, especially in 3 subgroups of patients who received surgery, TACE, and PEI
							AJCC	IV (253); II (193)	102.24	-	
							Okuda	I (375); II (278)	92.01	-	
Ueno et al. [53]	‘02	Japan	R	662	-	LR	CLIP 4th edition	III (275); II 223	155.61	-	CLIP score 4th edition has a higher stratification value than the 3rd edition. However, this benefit is due to the non-surgical patients, rather than to the surgical patients.
							CLIP 3th edition	IVA (237); II (196)	122.52	-	
Cillo et al. [46]	‘04	Italy	R	187	11 (0.3–120)	LR AIC	BCLC	B (43); C (25)	70.67	953.02	BCLC is superior in surgical and non surgical patients;
							CLIP	1 (61); 0 (55)	41.29	984.40	
							Okuda	I (98); II (79)	36.52	985.18	
							French	I.R. (92); L. R. (78)	34.88	986.82	
							CUPI	L.R.(157); I.R. [54]	27.49	994.21	
Kudo et al. [55]	‘04	Japan	R	4525	50 (14–156)	LR AIC	JIS	1 (1399); 2 (1471)	1238.05	33,642.3	JIS score performed better than CLIP score
							CLIP	1 (1687); 0 (1181)	1062.09	33,822.32	
Grieco et al. [56]	‘05	Italy	R	268	32 (3–130)	LR AIC	Okuda	I (190); II (78)	50.4	0.669	CLIP and BCLC more effective than in early-intermediate HCC in HCC underwent nonsurgical treatment or local treatments (PEI, RF, TACE). However, BCLC performed better in very early stage
							CLIP	0 (129); 1 (82)	76.8	0.726	
							BCLC	A4 (93); A2 (68)	89.9	0.731	
Toyoda et al. [57]	‘05	Japan	R	1508	-	AIC	JIS	1 (349); 2 (311)	-	9987.96	Comparison in two era: pre e post 1991. JIS system is the appropriate system in current era of early detection and treatment of HCC.
							CLIP	1 (396); 0 (332)	-	10,031.8	
							BCLC	A (632); C (418);	-	10,079.7	
Marrero et al. [44]	‘05	USA	R	244	-	LR AIC	BCLC	C (31); A (28)	76.8	943.7	The BCLC staging system provided the best prognostic stratification.
							GRETCH	B (42); C (39)	59.2	970.4	
							TNM	III (36); II (31)	54.3	978.5	
							Okuda	2 (44); 3 (37)	52.9	974.4	
							CUPI	I (44); H (37)	52.3	990.8	
							CLIP	1 (31); 0 (19)	51.9	981.5	
							JIS	1 (25); 2–4 (19)	49.7	994.0	
Cillo et al. [58]	‘06	Italy	P	195	25 (5–54)		BCLC	A (89); B (58)	43.01	885.98	Including patient treated with LT, BCLC classification showed a better prognostic ability
							UNOS-TNM	II (75); IV (54)	20.03	915.62	
							JIS	2 (70); 1 (57)	12.45	928.16	
							Okuda	I (117); II (71)	3.98	933.06	
							CLIP	1 (63); 2 (54)	4.17	938.10	
Nanashima et al. [59]	‘06	Japan	R	230	-	AIC	Modified JIS	-	-	634.3	For HCC after hepatic resection modified JIS score is the best predictor of prognosis
							JIS	-	-	635.8	
							Modified CLIP	-	-	634.8	
							CLIP	-	-	636.5	
							Japan TNM	-	-	637.4	
Chung et al. [60]	‘07	Japan	R	290	-	LR AIC	JIS	2 (98); 1 (80)	138.0	1635.6	The JIS score provided the best prognostic stratification in a Japanese cohort of HCC patients who were mainly diagnosed at early stages and treated with radical therapies.
							BCLC	A (131); B (63)	111.0	1661.0	
							Tokyo	2 (75); 3 (60)	108.0	1671.4	
Huo et al. [61]	‘07	Taiwan	R	-	-	LR AIC	CLIP + MELD	-	192	1471.2	The MELD-based CLIP and JIS staging systems have an improved predictive ability compared to the original system and are feasible models for HCC staging in the MELD era
							CLIP	-	173.5	1489.7	
							JIS + MELD	-	140.7	1522.5	
							BCLC + MELD	-	126.9	1536.3	
							JIS	-	124.7	1538.5	
							BCLC	-	122.9	1540.3	
Cho et al. [29]	‘07	Korea	R	131	24 (2–83)	LR AIC	CLIP	1 (55); 0 (34)	38.10	850.0	The CLIP system provided the best prognostic stratification for a cohort the patients with HCC who underwent TACE.
							JIS	2 (50); 1 (44)	33.6	854.5	
							Mod CLIP	1 (54); 2–0 [54]	26.9	863.2	
							Mod JIS	2 (51); 1 (46)	21.6	866.4	
							C-P score	6 (47); 5 (45)	18.8	869.2	
							Okuda	I (86); II (45)	13.5	868.5	
							BCLC	-	6.4	877.6	
Guglielmi et al. [62]	‘08	Italy	R	112	24 (3–92)	LR	BCLC	B (33); A4 (31)	15.1	-	BCLC performed better in HCC patients who underwent RF; moreover, it can give important prognostic information after complete response to treatment.
							GRETCH	A (60); B (34)	12.4	-	
							Okuda	I (63); II (28)	10.5	-	
							CUPI	LR (88); I (8)	8.0	-	
							JIS	2 (53); 1 (28)	3.9	-	
							CLIP	1 (46); 0 [54]	1.9	-	
							TNM	I (55); II (37)	1.3	-	
Chen et al. [63]	‘08	Taiwan	R	2010	-	LR AIC	Tokyo	-	279.1	19,383.7	The Tokyo score was the most informative one for predicting the survival of HCC patients as a whole, receiving LR, or TACE. CLIP score was the best fit system for HCC patients receiving CT or BSC. Each stagingsystem showed a significant difference in predicting the probability of survival across different stages. The applicability of staging systems for patients with HCC was dependent on treatment methods.
							JIS	-	213.3	19,492.3	
							CLIP	-	205.3	19,508.3	
							BCLC	-	191.2	19,540.1	
							Okuda	-	184.7	19,551.7	
							TNM	-	93.3	19,782.9	
Chen et al. [64]	‘07	Taiwan	R	382	21 (0.1–120)	LR	CLIP	-	131.3	-	While the CLIP system should be considered to stage major hepatectomy patients, the JIS system could be chosen to stage minor hepatectomy patients.
							JIS	-	122.8	-	
							BCLC	-	94.7	-	
							Okuda	-	81.3	-	
							CUPI	-	55.8	-	
							AJCC	-	50.5	-	
Nanashima et al. [65]	‘05	Japan	R	210	-	AIC	Modified CLIP score	1 (138); 2 (43)	-	425.9	The modified CLIP score showed the lowest AIC for DFS and OS in HCC underwent liver resection.
							CLIP score	1 (156); 2 (30)	-	427.9	
							JIS score	1 (114); 2 (66)	-	436.4	
							AJCC TNM stage	I (125); II (61)	-	441.3	
							Japan TNM stage	II (132); III (56)	-	438.5	
Yen et al. [66]	‘09	Taiwan	R	2882	-	AIC	CLIP	-	-	35,21	CLIP system provided the best prognostic stratification in late stages HCC. TNM-based JIS combined aFP may be the most applicable in early-stage HCC patients
							TNM-based JIS	-	-	35.42	
							TNM-based JIS + aFP	-	-	35.27	
							BCLC	-	-	35.58	
							JIS	-	-	35.47	
Chan et al. [67]	‘10	China	P	595	41.4 (40–46.6)	LR AIC	CLIP	-	213.05	5791.02	CUPI is an appropriate staging system for HBV-related HCC. In patients with advanced HCC, both CUPI and CLIP offer good risk stratification
							CUPI	-	197.04	5807.03	
							Okuda	-	154.57	5849.50	
							TNM	-	90.17	5913.91	
							BCLC	-	64.39	5939.68	
Chen et al. [63]	‘10	Taiwan	R	2010	-	LR AIC	Tokyo	-	552.2	19,383.7	The Tokyo staging system was the best in predicting survival for patients receiving LR or TACE while CLIP scoring system was the most suitable in predicting survival in HCC patients receiving CT or BSC
							JIS	-	443.5	19,492.3	
							CLIP	-	427.5	19,508.3	
							BCLC	-	395.8	19,540.1	
							Okuda	-	384.1	19,551.7	
							TNM	-	153.0	19,782.9	
Tournoux et al. [68]	‘11	France	P	416	48	AIC	Tournoux-Facon score	-	-	3884	The new prognostic score and CLIP + PS are recommended in palliative settings
							CLIP + PS	-	-	3894	
							CLIP	-	-	3906	
							GRETCH	-	-	3910	
							BCLC	-	-	3928	
							Okuda	-	-	3913	
Op de Winkel et al. [69]	‘12	Germany	R	405	14 (0.2–113)	LR AIC	CLIP	1 (131); 2 (80)	-	2286	CLIP-score was identified as the most suitable staging system for predicting prognosis in a large German cohort of predominantly non-surgical HCC-patients
							JIS	2 (135); 3 (85)	-	2293	
							Okuda	I (202); II (145)	-	2337	
							GETCH	Intermediate 176; Low 103	-	2342	
							TNM	I (122); III (114)	-	2342	
							BCLC	C (138); B (99)	-	2343	
							Child	A (130); B (120)	-	2369	
Gomaa et al. [70]	‘14	Egypt	P	2000	15 (13.6–16)	LR	BCLC	B (608); A (501)	810	-	BCLC staging system provided the best prognostic stratification for HCC patients. However, CLIP score has the highest stratification ability in patients with advanced HCC highlighting the importance of including aFP in best staging system.
							JIS	2 (625); 3 (579)	694	-	
							CLIP	2 (531); 1 (507)	679	-	
							Okuda	II (917); I (696)	363	-	
Memon et al. [71]	‘14	USA	R	428	23.2		CLIP	2 (115); 1 (113)	127.22	2992.80	CLIP was most accurate in predicting HCC survival in patients after Y-90 TARE treatment.
							JIS	4 (140); 3 (139)	103.98	3014.04	
							UNOS	T4b [54]; T2 (99)	94.61	3023.41	
							BCLC	C (196); B (122)	81.97	3032.05	
							GRETCH	B (239); A (95)	73.40	3038.61	
							CUPI	LR (347); IR (72)	64.45	3047.57	
							Okuda	2 (266); 1 (151)	53.13	3058.89	
							CTP	B 215; A 201	38.00	3074.02	
Adhoute et al. [32]	‘15	Taiwan	R	3182	17	LR AIC	HKLC	I (1001); II (862)	1370.15	5334.15	Compared with the BCLC system, the HKLC system has better prognostic accuracy
							BCLC	C (1282); A (736)	920.99	5582.84	
Yan et al. [72]	‘15	China	R	668	-	AIC	HKLC	I (267); IIb (201)	-	4709.48	Especially in HBV patient, the HKLC score is the best prognostic system in a Chinese cohort
							BCLC	A1-A2 (264);C (117)	-	4852.70	
Liu et al. [47]	‘16	Taiwan	R	3182	17	LR AIC	CLIP	-	1387.62	5666.83	CLIP score is the most accurate prognostic model
							TIS	-	1204.24	5782.58	
							HKLC	-	1078.52	5846.45	
							JIS	-	1058.81	5850.54	
							Tokyo	-	904.71	5966.41	
							BCLC	-	854.90	5973.09	
							French	-	874.57	5980.77	
							Okuda	-	841.81	6029.90	
							AJCC TNM-7	-	820.24	6035.86	
							CUPI	-	747.38	6100.00	
							TNM by LCSGJ		586.74	6201.56	
Farinati et al. [12]	‘16	Italy	R	5183	58 (26–106)	LR AIC	ITA.LI.CA	-	-	15,558	Comparison between 2003–2012 in internal and external validation(*n* = 3281). The ITA.LI.CA score showed the best discriminatory ability and monotonicity of gradients among the most common HCC staging systems
							CLIP	-	215.38	15,721	
							HKLC	-	179.83	15,728	
							MESIAH	-	285.23	15,772	
							JIS	-	356.03	15,898	
							Modified BCLC	-	411.35	15,952	
							BCLC	-	578.49	16,119	
Chen et al. [73]	‘17	China	R	220	-	LR AIC	CLIP	5 (57); 4 (50)	70.6	1601.5	CLIP performed better than others. CIS ranked second in predicting 3-month mortality
							CIS	2 (74); 1 (59)	48.4	1632.3	
							CUPI	1 (106); 2 (71)	46.7	1629.9	
							Okuda	II (119); III (67)	36.0	1641.1	
							TNM	III (127); IV (46)	21.0	1654.8	
							JIS	4 (83); 3 (56)	46.8	1627.4	
							BCLC	C (138); D (82)	7.24	1671.1	
Sohn et al. [74]	‘17	USA	R	1009	-	LR AIC	HKLC-9	-	250	6200	HKLC system determined prognosis in patients following TACE
							HKLC-5	-	201	6241	
							BCLC	-	119	6321	
Selby et al. [75]	‘17	Singapore	R	766	-	AIC	HKLC	I (208); II (203)	-	5711	HKLC has better performance in guiding treatment.
							BCLC	A (275); C (222)	-	5764	
Samawi et al. [76]	‘18	Canada	R	681	37.6 (29.5–41)	LR AIC	CLIP	2 (215); 1 (163)	63.37	5725.76	CLIP performed better while BCLC and TNM7 performed less favourably but the differences were small
							Okuda	1 (364); 2 (272)	50.76	5730.38	
							ALBI	2 (503); 1 (119)	24.40	5756.73	
							BCLC	C (591); B (37)	23.88	5759.25	
							TNM7	IV (394); III (148)	11.63	5771.51	
Borzio et al. [33]	‘18	Italu	R	1508	44 (23–63)	LR AIC	ITA.LI.CA	2 (299); 1 (270)	763	7087	The ITA.LI.CA system performed better than other multidimensional prognostic systems, even after stratification by curative or palliative treatment. This new system appears to be particularly useful for predicting individual HCC prognosis in clinical practice.
							CLIP	0 (619); 1 (422)	575	7233	
							HKLC	I (608); IIa (304)	659	7194	
							MESIAH	Q3 (380); Q4 (377)	642	7159	
							JIS	1 (583); 2 (326)	563	7206	
							ITA.LI.CA tumor stage	-	482	7307	
							BCLC	A (687); B (310)	514	7234	

aFP: Alfa Feto-Protein; AIC: Akaike information criterion; ALBI: ALbumin-BIlirubin score; AJCC: American Joint Committee on Cancer; BCLC: Barcelona Clinic Liver Cancer staging classification; BSC: Best Supportive Care; CLIP: Cancer of the Liver Italian Program Score; CT: Chemo Therapy; DFS: Disease Free Survival; GRETCH: GRoupe d’Etude et de Traitement du Carcinome Hépatocellulaire; HKLC: Hong Kong Liver Cancer Staging; JIS: Japanese Integrated Staging score; I.R.: Intermediate Risk; ITA.LI.CA: ITAlian LIver Cancer; L.R.: low risk; LR: likelihood ratio; LR: Liver Resection; LT: Liver Transplantation; P: Prospective; PEI: Percutaneous Ethanol Injections; PS: Performance Status; R: Retrospective; RF: Radio Frequency; OS: Overall Survival; TACE Transcatheter Arterial ChemoEmbolizations; TARE: Trans Arterial RadioEmbolization; TNM: Classification of Malignant Tumors; UNOS: United Network for Organ Sharing; V: value.

**Table 7 cancers-13-01673-t007:** Discrimination ability, monotonicity of gradient, and homogeneity of different HCC prognostic systems.

HCC Prognostic Systems	C index	Trend χ^2^ Test	AIC	LR Test, *p* Value
Study Group (*n* = 6882)				
ITA.LI.CA score [35]	0.693	1749	74,138	-
ITA.LI.CA staging [35]	0.667	1211	74,520	472 < 0.0001
MESH [8]	0.662	1182	74,476	518 < 0.0001
CNLC [9,10,11]	0.661	1187	74,555	522 < 0.0001
BCLC [15]	0.659	1114	74,558	537 < 0.0001
CLIP [14]	0.620	1143	74,719	882 < 0.0001

In the columns are reported the C-index, the test for trend chi-square and the AIC of the tested prognostic models. The higher the C-index and the test for trend chi-square, the higher the discriminatory ability and monotonicity of gradients. The lower the AIC value, the higher the homogeneity and the monotonicity of gradients. In addition, in the last column the ITA.LI.CA score was compared with other systems by using the likelihood ratio test: the higher the test value, the higher the superiority of the ITA.LI.CA score over the compared system. Abbreviations: C, concordance; χ^2^, chi square; AIC, Akaike Information Criterion; LR, likelihood ratio; ITA.LI.CA, Italian Liver Cancer; CNLC, Chinese Liver Cancer; BCLC, Barcelona Clinic Liver Cancer; CLIP, Cancer Liver Italian Program.

## Data Availability

The authors confirm that the data supporting the findings of this study are available within the article and its Appendix A.

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
