# Peer review of "Overview of Prognostic Systems for Hepatocellular Carcinoma and ITA.LI.CA External Validation of MESH and CNLC Classifications"

_cancers, 2021, doi:10.3390/cancers13071673_

Round 1

Reviewer 1 Report

The authors present an excellent manuscript with an overview of available prognostic systems, an overview of the methodology for comparing different staging systems and a detailed analysis of the main published scores.

In addition they present a formal comparison between the ITA.LI.CA score and staging system and MESH, CNLC, BCLC and CLIP score demonstrating the superiority of the ITA.LI.CA score in an extremely large Western cohort.

The authors must be commended for their excellent work.

There are only several minor issues that should be addressed prior to publication:

Overview:

  • P4: Even this system does not propose treatment recommendations and has not been externally validated in Western countries.
    • There is at least one publication with a validation of the MESH score in a Western country (Heinrich et al, Validation of prognostic accuracy of MESH, HKLC, and BCLC classifications in a large German cohort of hepatocellular carcinoma patients, United European Gastroenterol J . 2020 May;8(4):444-452.

doi: 10.1177/2050640620904524. Epub 2020 Jan 29.

  • Please comment

Language:

  • Overall, the manuscript is very well written and comprehensible. There are only minor mistakes which should be corrected prior to publication, e.g.
    • P11: Formatting is off in the first column, first row
    • Alpha-Fetoprotein is sometimes abbreviated as AFP and sometimes as aFP, a homogenous presentation would be beneficial
    • P16: In column are repoted C-index, the test for trend chi-square and the AIC of the tested prognostic models.
      • Sentence is difficult to understand. Is repoted correct?

Author Response

The authors present an excellent manuscript with an overview of available prognostic systems, an overview of the methodology for comparing different staging systems and a detailed analysis of the main published scores.

In addition they present a formal comparison between the ITA.LI.CA score and staging system and MESH, CNLC, BCLC and CLIP score demonstrating the superiority of the ITA.LI.CA score in an extremely large Western cohort.

The authors must be commended for their excellent work.

There are only several minor issues that should be addressed prior to publication:

Overview:

  • P4: Even this system does not propose treatment recommendations and has not been externally validated in Western countries, there is at least one publication with a validation of the MESH score in a Western country (Heinrich et al, Validation of prognostic accuracy of MESH, HKLC, and BCLC classifications in a large German cohort of hepatocellular carcinoma patients, United European Gastroenterol J . 2020 May;8(4):444-452.

ANSWER: We thank the reviewer for these positive comments. We reported the suggested reference in the revised paper changing also the relative sentence: “Even this system does not propose treatment recommendations. However, MESH score had at least on external validation in European/North American countries (doi: 10.1177/2050640620904524. Epub 2020 Jan 29.)

Language:

  • Overall, the manuscript is very well written and comprehensible. There are only minor mistakes which should be corrected prior to publication, e.g.
    • P11: Formatting is off in the first column, first row:
    • Alpha-Fetoprotein is sometimes abbreviated as AFP and sometimes as aFP, a homogenous presentation would be beneficial:
    • P16: In column are repoted C-index, the test for trend chi-square and the AIC of the tested prognostic models. Sentence is difficult to understand. Is repoted correct?

ANSWER: As suggested, we formatted the first row of the table at P11. The AFP was changed to aFP. Finally, we changed according to your suggestion the phrase in P16: “In columns are reported C-index, the test for trend chi-square and the AIC of the tested prognostic models”

Reviewer 2 Report

The authors describe first an overview of the prognostic systems for HCC, second, the methodology required for a correct comparison between different systems in terms of prognostic performance, and, lastly, a comparison between the prognostic systems using a large Italian database including 6,882 HCC patients.

The manuscript is interesting. I have only a few minor comments.

Minor comments

#1. P2, line 40. “East” may be “Western”.

#2. Alpha-fetoprotein are abbreviated as both aFP and AFP.

#3. P5, line 35. “Figure 2” may be “Table 3”.

Author Response

Answer to Reviewer 2

The authors describe first an overview of the prognostic systems for HCC, second, the methodology required for a correct comparison between different systems in terms of prognostic performance, and, lastly, a comparison between the prognostic systems using a large Italian database including 6,882 HCC patients.

The manuscript is interesting. I have only a few minor comments.

#1. P2, line 40. “East” may be “Western”.

#2. Alpha-fetoprotein are abbreviated as both aFP and AFP.

#3. P5, line 35. “Figure 2” may be “Table 3”.

ANSWER: We thank the reviewer for these positive comments. As suggested, in P2 line 40 we changed Eastern and Western to Asian and European/North American to be more specific. The AFP was changed to aFP throughout the paper. Finally, Figure 2 was changed to Table 3 as suggested.

Reviewer 3 Report

Thank you for the opportunity to review the manuscript by members of the AISF HCC Special interest group and Italian Liver Cancer (IT.LI.CA) study group, which provides an overview of prognostic classification tools devised and used for patients with hepatocellular carcinoma. The correlation of radiographic and morphological features with physiological sequelae has been attempted by numerous researchers and hepatobiliary surgeons, since the degree to which the two interact profoundly affects prognosis, disease severity, and morbidity secondary to therapy.

The authors then provide an external validation of the MESH and CNLC scores using the ITA.LI.CA database and compare the performance of these scores to previously studied and discussed frameworks with several diagnostic measures including AIC, Harrell’s C index, and likelihood ratio tests.

Major comments

The authors highlight the motivations for such an overview throughout their manuscript, and this type of summary is helpful for clinicians and researchers who otherwise work mainly with a small subset of these clinical prediction models. Comparing the performance of each score is helpful when confronted with contrasting predictions in clinical practice. This is a strength. The main criticisms given below fall into two categories: (1) the conceptual organization of the review and (1) the appropriate demonstration of comparisons

1a) The authors’ conceit of “data-based”, “evidence-based”, and “combined” is confusing. The weighting of expert opinion versus the inclusion of data related to physiology versus tumor characteristics clearly varies from score to score, but I’m not convinced that this categorization is clear or intuitive. I defer to other more experienced clinicians about the proposed categorization the scores for expository purposes, but for the average reader this organization lacks clarity.

1b) While it is helpful for a brief narrative to accompany the table highlighting the relative strengths and weaknesses of the score, the authors appear to list these in sequential paragraphs. Rather, a summary paragraph drawing attention to particularities of a set of scores would be preferable (e.g. along the lines of “the most/least accurate scores lack convincing data of external validity”).

2a) The relative performance of different scores is, as above, difficult to compare across categories that are affected by different treatments. “Recategorizing” patients under a different scoring system to determine how it performs relative to others is still subject to bias introduced by the score that guided the patient’s actual treatment. This concern is all the greater given that the discriminatory ability of the scores being validated diminishes for higher stage cancers. This needs to be addressed.

2b) Could the authors indicate whether treatment for patients using the reference would have changed under the scoring system being validated? If so, could the authors provide a sensitivity analysis examining comparisons of patients re-categorized but for whom treatment would NOT have  changed?

Minor considerations

  • While parlance of “Eastern” and “Western” is used widely with respect to HCC epidemiology, it is hardly specific. I’d recommend shifting these references to the continent of origin (Asian and European/North American) if the authors believe it is useful to identify commonalities that way (see reference).
  • The statement “patient survival is accurately predicted, at least in the training population” is tautological. Rather, the authors should indicate the extent to which these models are derived from a process that is agnostic to known risk factors (i.e., was the relevance and contribution of physiologic metrics tested in order to be included, and if so, how?)
  • Avoid using words such as “strong” to describe statistical methodology or results.
  • The phrase “a pure prognostic point of view” is vague when describing the BCLC, as the following sentence clarifies its performance relative to other scores.
  • Section 4.2: “Survival curves were also stratified according to stages, and the log rank test was used to compare differences in survival.” The use of “stratified” is not clear. Were there cross-score comparisons? Or were these all within-score?

Choo SP, Tan WL, Goh BKP, Tai WM, Zhu AX. Comparison of hepatocellular carcinoma in Eastern versus Western populations. Cancer. 2016 Nov 15;122(22):3430-3446. doi: 10.1002/cncr.30237. Epub 2016 Sep 13. PMID: 27622302.

Finally, incidentally discovered typographical and grammatical errors are listed below. These are not meant to convey judgement of the scientific merit of the manuscript but are simply noted to help the authors more efficiently revise their paper.

  • There are multiple instances of inconsistent spacing after periods. Please ensure consistency.
  • There are also multiple instances of phrasing that includes unusual constructions or word choices
  • Simple summary: “in order to give a contribute to this topic” should be rephrased, as “both waiting for this kind of validation”
  • Introduction, paragraph 2: “…usually rises” should be “usually arises”
  • Harrel C index should be Harrell’s C-index
  • Page 4: “lacks of a strong” should be “lacks a …”

Author Response

Answer to Reviewer 3

Thank you for the opportunity to review the manuscript by members of the AISF HCC Special interest group and Italian Liver Cancer (IT.LI.CA) study group, which provides an overview of prognostic classification tools devised and used for patients with hepatocellular carcinoma. The correlation of radiographic and morphological features with physiological sequelae has been attempted by numerous researchers and hepatobiliary surgeons, since the degree to which the two interact profoundly affects prognosis, disease severity, and morbidity secondary to therapy.

The authors then provide an external validation of the MESH and CNLC scores using the ITA.LI.CA database and compare the performance of these scores to previously studied and discussed frameworks with several diagnostic measures including AIC, Harrell’s C index, and likelihood ratio tests.

Major comments

The authors highlight the motivations for such an overview throughout their manuscript, and this type of summary is helpful for clinicians and researchers who otherwise work mainly with a small subset of these clinical prediction models. Comparing the performance of each score is helpful when confronted with contrasting predictions in clinical practice. This is a strength. The main criticisms given below fall into two categories: (1) the conceptual organization of the review and (2) the appropriate demonstration of comparisons

ANSWER: We thank the reviewer for his/her positive comments, but above all for his/her criticisms giving us the possibility to considerably improve our paper.

1a) The authors’ conceit of “data-based”, “evidence-based”, and “combined” is confusing. The weighting of expert opinion versus the inclusion of data related to physiology versus tumor characteristics clearly varies from score to score, but I’m not convinced that this categorization is clear or intuitive. I defer to other more experienced clinicians about the proposed categorization the scores for expository purposes, but for the average reader this organization lacks clarity.

1b) While it is helpful for a brief narrative to accompany the table highlighting the relative strengths and weaknesses of the score, the authors appear to list these in sequential paragraphs. Rather, a summary paragraph drawing attention to particularities of a set of scores would be preferable (e.g. along the lines of “the most/least accurate scores lack convincing data of external validity”).

ANSWER: We thank the reviewer for these comments. Considering your suggestions, we changed the conceit of “data-based” and “evidence-based” with “Prognostic scores” and “Staging systems” that should be less confusing for the reader. In brief, the concept of prognostic systems includes both prognostic score (previously defined “data-based systems”), staging systems (previously defined “evidence-based systems), and combined systems (systems able to be used both as a score and as a staging system). Furthermore, we created a new paragraph called “Summary of pro and cons of prognostic systems categories” at P6, outlining the pro and cons of each prognostic systems category.

2a) The relative performance of different scores is, as above, difficult to compare across categories that are affected by different treatments. “Recategorizing” patients under a different scoring system to determine how it performs relative to others is still subject to bias introduced by the score that guided the patient’s actual treatment. This concern is all the greater given that the discriminatory ability of the scores being validated diminishes for higher stage cancers. This needs to be addressed.

2b) Could the authors indicate whether treatment for patients using the reference would have changed under the scoring system being validated? If so, could the authors provide a sensitivity analysis examining comparisons of patients re-categorized but for whom treatment would NOT have  changed?

ANSWER: We thank the reviewer for these comments giving us the possibility to elaborate on the relevant issue concerning the relationship between prognostic systems and treatment choice. This complex relationship can be analyzed from two points of view (corresponding to reviewer’s comments 2a and 2b, respectively).

The first is mainly a prognostic point of view (answer to comment 2a). Should we consider treatment as an independent variable to be included in a prognostic system? Or should we develop different prognostic scores for each treatment procedure? Since, treatment selection is influenced by different prognostic variables (i.e. tumour characteristics, liver function, and patient general conditions) there is a statistical interaction between treatment and other variables, so treatment can not be included as an additive variable in a general prognostic system.

From this specific prognostic point of view, therefore, commonly used prognostic systems (described in this paper) can be used for a prognostic assessment for the general HCC population, but specific prognostic scores for each treatment should be used to obtain a more accurate prognostic estimation after that treatment decision is taken.

So to specifically answer to reviewer’s comment 2a, in this review we only described prognostic systems designed for a general HCC population independently from treatment choice, while treatment specific prognostic scores are not objects of this study.

The second point of view concerns the relationship between prognostic systems and treatment assignment. As described in this paper, only staging and combined systems categories proposed treatment algorithms for HCC patients. Although it is not a specific object of this study, we could comment that any attempt to link treatment choice to a specific stage and the use of strict algorithms was not successful in clinical practice. Several evidences from the literature showed, in fact, that adherence to guidelines was very low in every day clinical practice (see new references 37-40). We sincerely think that multidisciplinary evaluation aimed to obtain a personalized treatment decision is the best way to optimize HCC patient outcome. On this perspective, the treatment hierarchy approach is probably closer than stage hierarchy to precision medicine therapeutic approach for HCC. We added some of these comments in the new “Pro and cons paragraph”.

Minor considerations

  • While parlance of “Eastern” and “Western” is used widely with respect to HCC epidemiology, it is hardly specific. I’d recommend shifting these references to the continent of origin (Asian and European/North American) if the authors believe it is useful to identify commonalities that way (see reference).
  •  

ANSWER: We thank the reviewer for this comment. In particular, as suggested, we specified in the manuscript Eastern and Western to more specific Asian and European/North American populations. Furthermore, we added the suggested reference (10.1002/cncr.30237).

  • The statement “patient survival is accurately predicted, at least in the training population” is tautological. Rather, the authors should indicate the extent to which these models are derived from a process that is agnostic to known risk factors (i.e., was the relevance and contribution of physiologic metrics tested in order to be included, and if so, how?)

ANSWER: We thank the reviewer for this comment. Considering it, we changed the statement “patient survival is accurately predicted, at least in the training population” to “These systems rely on a rigorous statistical methodology usually based on multivariable survival models derived from a process that is agnostic to known risk factors.”

  • Avoid using words such as “strong” to describe statistical methodology or results.

ANSWER: As suggested, we changed the word strong, changing the phrase at P7: “However, they are based on a statistical methodology that can be debated and often lack prognostic power”

  • The phrase “a pure prognostic point of view” is vague when describing the BCLC, as the following sentence clarifies its performance relative to other scores.

ANSWER: As suggested, we changed the phrase at P5: “The BCLC suffers from the fact that it was not created and weighted…”

  • Section 4.2: “Survival curves were also stratified according to stages, and the log rank test was used to compare differences in survival.” The use of “stratified” is not clear. Were there cross-score comparisons? Or were these all within-score?

ANSWER: We thank the reviewer for these comments. According to your suggestion, we changed to: “Kaplan Meier curves were used to describe survival figures of different stages of each prognostic system, and the log rank test was used to compare differences in survival.”

Finally, incidentally discovered typographical and grammatical errors are listed below. These are not meant to convey judgement of the scientific merit of the manuscript but are simply noted to help the authors more efficiently revise their paper.

  • There are multiple instances of inconsistent spacing after periods.
  • There are also multiple instances of phrasing that includes unusual constructions or word choices
  • Simple summary: “in order to give a contribute to this topic” should be rephrased, as “both waiting for this kind of validation”
  • Introduction, paragraph 2: “…usually rises” should be “usually arises”:
  • Harrel C index should be Harrell’s C-index:
  • Page 4: “lacks of a strong” should be “lacks a …”

ANSWER: We thank the reviewer for these comments. We changed and reviewed the paper according to your suggestions. The changes have been tracked in the paper.

Round 2

Reviewer 3 Report

The authors have taken care to address the comments on their prior submission and suitably provided revisions that have both improved clarity of their overall message and the manuscript's readability

No further concerns at this time.